# Inpatient morbidity and mortality of measles in the United States

Raj Chovatiya[1], Jonathan I. Silverberg[2]*

**1** Department of Dermatology, Northwestern University Feinberg School of Medicine, Chicago, IL, United States of America, **2** Department of Dermatology, The George Washington University School of Medicine and Health Sciences, Washington, DC, United States of America

\* JonathanISilverberg@Gmail.com

## Abstract

### Background

Measles is an extremely contagious, vaccine-preventable infection that was officially declared eradicated in the US in 2000. However, measles outbreaks are increasingly occurring in the US. Measles cases have considerable morbidity requiring hospitalization, yet little is known about hospitalization and complications from measles in recent years.

### Objectives

To analyze the frequency, predictors, costs and other outcomes of hospitalization for measles in the US.

### Methods

The 2002–2016 Nationwide Inpatient Sample, containing a 20% sample of US hospitalizations (n = 96,568,625), was analyzed. Measles and comorbidities were defined by International Classification of Disease, Ninth Revision, Clinical Modification (ICD-9-CM) or ICD-10-CM codes. Multivariable survey logistic regression and linear regression models controlling for sociodemographic demographic factors were constructed to understand associations with organ-specific complications, and cost of care and length of stay, respectively.

### Results

Overall, 1,018 measles hospitalizations occurred in 2002–2016, and hospitalizations increased over time. In multivariable logistic regression models, measles was associated with higher odds of gastrointestinal, hematologic, infectious, neurologic, ophthalmologic, pulmonary, and renal complications, with the strongest association observed with encephalitis (39.84 [16.51–96.12], P<0.0001). Increased length of stay (LOS) and similar cost of care (mean [95% CI]; 4.8 [4.4–5.4]; $7,438 [$6,446-$8,582]) were observed versus (vs.) all other admissions (4.5 [4.4–4.5]; P<0.01; $7,854 [$7,774-$7,935], P>0.05). There were 34 deaths in hospitalized measles patients; inpatient mortality was numerically higher in those with vs. without measles (proportion ± SEM: 3.3±1.2% vs. 2.3±0.01%, P = 0.333).

**Data Availability Statement:** All relevant data are within the manuscript and its Supporting Information files.

**Funding:** This publication was made possible with support from the Agency for Healthcare Research and Quality (AHRQ) (JIS), the Dermatology

Foundation (JIS), NIH K12 HS023011 (JIS), and NIH T32 AR060710 (RC). The funders had no role in study design, data collection and analysis, decision to publish, or preparation of the manuscript.

**Competing interests:** The authors have declared that no competing interests exist.

**Abbreviations:** ICD-9-CM, International Classification of Disease 9th edition Clinical Modification; ICD-10-CM, International Classification of Disease 10th edition Clinical Modification; NIS, Nationwide Inpatient Sample; HCUP, Healthcare Cost and Utilization Project; CDC, Centers of the Disease Control and Prevention.

## Limitations

Lack of outpatient or prescription data.

## Conclusions

Measles continues to pose a substantial and preventable health care burden, with serious complications, hospitalization and inpatient mortality. Further studies are needed to improve the prevention and management of measles.

## Introduction

Measles is a highly contagious and potentially life-threatening, airborne disease characterized by high fever, cough, coryza, conjunctivitis, and morbilliform rash. Measles can cause numerous organ-specific complications that may lead to inpatient hospitalization and even death, including gastrointestinal [1–3], neurologic [4–9], pulmonary [10–13], ophthalmologic [14–16], hematologic [17], renal [18], and dermatologic complications [19].

Prior to the introduction of measles vaccination in 1963, there were >100 million measles cases resulting in 6 million deaths worldwide, with 4 million cases and 450 deaths in the US annually [20]. Despite major strides in vaccine coverage, measles is still a leading cause of vaccine-preventable death, especially in children, with more than 20 million new cases and 100,000 deaths worldwide annually [21,22]. In the US, due to a highly successful public health campaign based on universal vaccination, measles was officially declared eliminated (i.e., absence of endemic transmission for ≥12 months) in 2000 [23,24]. However, the US has faced a resurgence of measles outbreaks in recent years, driven largely by travel-related exposures and communities with low rates of vaccination [25]. According to the Centers of the Disease Control and Prevention (CDC), there were 1,282 confirmed cases of measles in 31 states with 128 hospitalizations from January to December 2019, the highest yearly total since the year 1992; this trend that has been mirrored worldwide.

Severe cases of measles require hospitalization. Based on historical data, the CDC has estimated that approximately 1 in 4 of cases of measles in the US result in hospitalization, and 1 in 1000 cases results in death. Hospitalizations for measles precipitously declined with widespread measles vaccination [26,27]. Without vaccination, there would be 400,000 hospitalizations costing >$3 billion USD and >1,800 deaths annually [27,28]. However, few studies examined the occurrence of complications, hospitalization, and mortality secondary to measles since its resurgence in the US. This study sought to analyze the frequency, predictors, costs and other outcomes of hospitalization for measles in the US.

## Materials and methods

The 2002–2016 Nationwide Inpatient Sample (NIS) was analyzed. The NIS is sponsored by the Healthcare Cost and Utilization Project (HCUP) of the Agency for Healthcare Research and Quality (http://www.hcup-us.ahrq.gov). Each year of the NIS contains an approximately 20% stratified representative sample of all hospitalizations in the US (e.g. general, specialty, academic, children's, etc.). The hospitals are stratified by ownership/control, bed size, teaching status, urban/rural location, and the nine U.S. census divisions. A systematic sampling system utilizing a self-weighted sample design is used to draw a sample of discharges from all hospitals that is representative of the US population based on the following factors: de-unidentified hospital number,

census division of hospital, hospital ownership, urban-rural location of hospital (ranging from micropolitan to metropolitan areas), hospital teaching status, number of beds in the hospital, diagnosis-related group for the hospital stay, and admission month of the hospital stay. This large sample size allows for analyses of uncommon conditions and highly specific patient populations. Sample weights are created by the NIS that factor in the study sampling design to allow for representative estimates of hospital discharges across the US and are determined by the ratio of universe discharges (based on births and admission) to sampled discharges within a specific stratum. The NIS does not include inpatient laboratory or treatment data. All data were de-identified, and no attempts were made to identify any of the individuals in the database. Patient consent was not obtained because the databases were received de-identified. All parties with access to the HCUP were compliant with the HCUP's formal data use agreement. This study was approved by the institutional review board at Northwestern University.

The NIS lists one primary diagnosis and up to 24 secondary diagnoses. The databases were searched for a primary or secondary diagnosis of measles and complications using International Classification of Disease, Ninth Revision, Clinical Modification (ICD-9-CM) or ICD-10-CM codes (S1 Table). Diagnoses for complications were selected based on previous associations in the literature or to explore for novel associations.

To control for readmitted cases that may be counted as unique encounters in the NIS, we excluded patients who were transferred to short-term hospitals with planned readmission (4.8% of measles cases in the cohort).

## Data analysis

Statistical analysis was performed using survey procedures adjusting for sample weighting, clustering, and strata in SAS version 9.4 (SAS Institute, Cary, NC). Baseline characteristics of inpatients with and without a measles diagnosis were determined. The cost for inpatient care was calculated based on the total charge of the hospitalization and the cost-to-charge ratio estimated by HCUP. Costs were adjusted for inflation to the year 2018 according to the Consumer Price Index from the US Bureau of Labor Statistics. Weighted t-tests and Rao-Scott chi-square tests compared the characteristics of mean and categorical variables, respectively.

Associations of measles hospitalization were examined including age, number of chronic conditions (defined as lasting $\geq$ 12 months and meeting criteria regarding limitations on independence and/or need for ongoing care), discharge quarter, sex, health insurance coverage (including Medicare, a federally-funded program primarily for those $\geq$65 years, and Medicaid, a means-tested, state and federally-funded program for those with low income), hospital location, race/ethnicity, median annual income of the hospital zip code, and year. Multivariable logistic regression models were constructed with measles diagnosis as the independent variable (yes/no) and various complications (yes/no) as the dependent variables. Multivariable linear regression models were constructed with measles diagnosis as the independent variable (yes/no), and log-transformed cost or LOS as the dependent variables (continuous). Cost and LOS were log-transformed because model residuals were not normally distributed for the untransformed variables. Complete case-analysis was performed. Post-hoc correction for multiple dependent tests was performed by minimizing the false discovery rate with the approach of Benjamini and Hochberg [29]. Two-sided, corrected P-values $\leq$0.05 were considered statistically significant.

## Results

### Population characteristics

There were 96,568,625 discharges (weighted frequency: 466,712,770) analyzed in the NIS between 2002 and 2016, including 1,018 weighted cases of hospitalization for measles. The

estimated incidence of measles hospitalizations was 2.2 per ten million persons [ptm] and increased between 2002 and 2016 (Fig 1A).

The mean ± standard error of the mean (SEM) age of measles inpatients was 32.0 ± 1.9 years, with a majority being <10 and ≥40 years (32.0% and 41.3%) and male (56.1%) (Table 1). Of the measles inpatients <10 years of age, 64.5% were 0–1 year, 25.1% were 2–5 years, and 10.4% were 6–9 years. The incidence of hospitalization decreased with age (0–9 years: 5.4 ptm; ≥40 years: 2.0 ptm) (Fig 1B). Most patients were healthy at baseline with ≤1 chronic comorbid condition (51.8%) and presented to hospital in a metropolitan area with population ≥ 1 million (62.9%). Private insurance (39.6%) and Medicaid (28.5%) were the most common payment sources. Inpatients with vs. without measles were significantly more likely to have non-white race/ethnicity overall (44.4% vs. 31.3%), including Hispanics (18.3% vs. 10.7%), and Asians/Pacific Islanders (9.0% vs. 2.3%). There were no significant differences in discharge season and income quartile in patients with vs. without measles.

## Factors associated with hospitalization for measles

Hospitalization for measles was associated with male sex (survey logistic regression: adjusted OR [95% CI]: 1.48 [1.27–1.72]) compared with females, Medicaid (1.29 [1.06–1.55]) or no insurance (1.64 [1.31–2.06]) compared with private, and Asian/Pacific Islander (3.24 [2.45–4.29]), Hispanic (1.35 [1.10–1.67]), or Native American/other (2.12 [1.62–2.77]) race/ethnicity compared with whites. Demographic factors inversely associated with measles hospitalization included increasing age (≥20 years: 0.38 [0.32–0.46] compared to 0–19 years), increasing number of chronic comorbid conditions (≥2: 0.57 [0.48–0.69] compared to ≤1), and decreasing

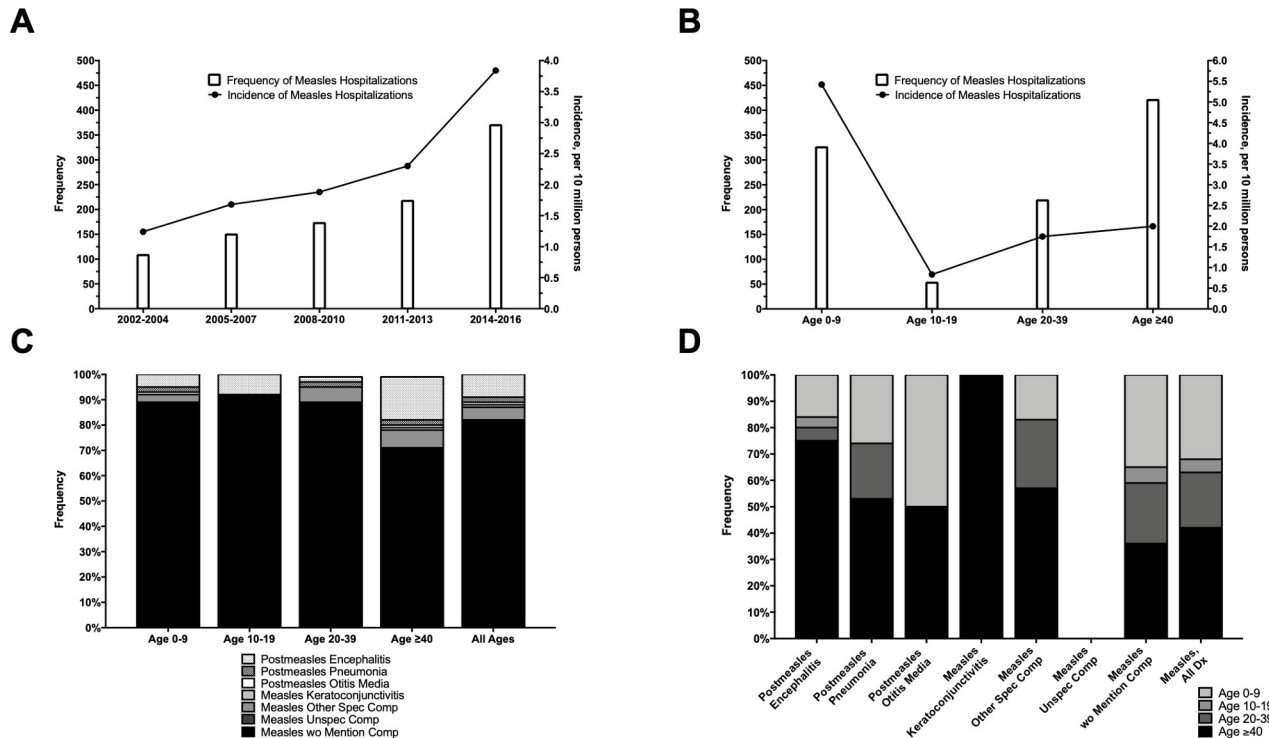

**Fig 1. Frequency of measles hospitalizations.** Measles hospitalizations by year (A), age (B), age stratified by ICD code (C), and ICD code stratified by age (D).

**Table 1. Baseline characteristics of inpatients with and without measles.**

| Variable | Measles | | |
|---|---|---|---|
| | **No** | **Yes** | **P-value** |
| **Age–mean (SEM)** | 54.6 (0.1) | 32.0 (1.9) | <0.0001 |
| **Age–wtd freq (%)** | | | |
| 0–9 | 34,914,571 (7.5%) | 325 (32.0%) | <0.0001 |
| 10–19 | 14,718,915 (3.2%) | 53 (5.2%) | |
| 20–39 | 71,217,494 (15.3%) | 219 (21.5%) | |
| ≥40 | 345,466,299 (74.1%) | 421 (41.1%) | |
| **Sex–wtd freq (%)** | | | |
| Male | 208,835,843 (44.8%) | 571 (56.1%) | 0.001 |
| Female | 256,996,879 (55.2%) | 447 (43.9%) | |
| **Chronic Conditions–wtd freq (%)** | | | |
| 0–1 | 87,270,795 (20.7%) | 460 (51.8%) | <0.0001 |
| 2–5 | 185,504,649 (44.0%) | 296 (33.4%) | |
| ≥6 | 149,220,170 (35.3%) | 132 (14.9%) | |
| **Died–wtd freq (%)** | | | |
| No | 455,483,562 (97.7%) | 984 (96.7%) | 0.333 |
| Yes | 107,88,277 (2.3%) | 34 (3.3%) | |
| **Discharge Quarter–wtd freq (%)** | | | |
| Jan-Mar | 119,003,074 (25.5%) | 271 (26.8%) | 0.379 |
| Apr-Jun | 116,383,379 (25.0%) | 294 (29.0%) | |
| Jul-Sep | 115,217,670 (24.7%) | 220 (21.7%) | |
| Oct-Dec | 115,644,962 (24.8%) | 228 (22.5%) | |
| **Hospital Location–wtd freq (%)** | | | |
| Metropolitan ≥ 1 Million | 230,781,973 (53.4%) | 593 (62.9%) | 0.021 |
| Metropolitan < 1 Million | 121,268,896 (28.0%) | 233 (24.8%) | |
| Micropolitan | 47,133,884 (10.9%) | 82 (8.7%) | |
| Not Metropolitan or Micropolitan | 33,157,800 (7.7%) | 34 (3.7%) | |
| **Income quartile–wtd freq (%)** | | | |
| 1st | 127,041,374 (29.7%) | 243 (26.2%) | 0.472 |
| 2nd | 111,244,640 (26.0%) | 247 (26.6%) | |
| 3rd | 101,120,717 (23.6%) | 208 (22.5%) | |
| 4th | 88,357,189 (20.7%) | 230 (24.8%) | |
| **Primary Payer–wtd freq (%)** | | | |
| Medicare | 209,457,726 (45.0%) | 179 (17.6%) | <0.0001 |
| Medicaid | 75,330,911 (16.2%) | 290 (28.5%) | |
| Private insurance | 140,036,817 (30.1%) | 403 (39.6%) | |
| Self-Pay / No Charge | 25,579,210 (5.5%) | 121 (11.9%) | |
| Other | 15,433,769 (3.3%) | 24 (2.4%) | |
| **Race/Ethnicity–wtd freq (%)** | | | |
| White | 269,191,673 (68.7%) | 510 (55.6%) | <0.0001 |
| Black | 57,304,804 (14.6%) | 88 (9.6%) | |
| Hispanic | 42,076,891 (10.7%) | 168 (18.3%) | |
| Asian/Pacific Islander | 9,051,815 (2.3%) | 83 (9.0%) | |
| Native American | 2,360,698 (0.6%) | ≤10 (0.6%) | |
| Other | 12,095,380 (3.1%) | 63 (6.9%) | |

urban population (small metropolitan/micropolitan: 0.78 [0.67–0.92]; non-metropolitan/non-micropolitan: 0.51 [0.34–0.78] compared to metropolitan areas with ≥1 million population).

## Complications of measles hospitalization

Across all ages, the majority of measles inpatients were diagnosed as measles without mention of complications (82.1%) (Fig 1C). Diagnoses of measles with various specified complications, including encephalitis, pneumonia, and keratoconjunctivitis, generally had higher prevalence with increasing age, while measles without mention of complications was diagnosed more commonly in younger age (Fig 1D).

The most frequent complications observed with measles were: dehydration (weighted frequency [%]: 161 [15.8%]), hyponatremia (145 [14.3%]), pneumonia (127 [12.5%]), acute renal failure (106 [10.4%]), diarrhea (97 [9.5%]), thrombocytopenia (97 [9.5%]), conjunctivitis (87 [8.5%]), septicemia (84 [8.3%]), fever (73 [7.2%]), sepsis/SIRS (64 [6.3%]), bronchitis (49 [4.8%]), pleurisy (38 [3.8%]), otitis media (37 [3.7%]), and pancytopenia (35 [3.4%]) (Fig 2). Even among inpatients who were diagnosed as measles without mention of complications, 15.7% had dehydration, 14.5% had hyponatremia, 13.4% had pneumonia, 10.5% had diarrhea, 9.9% had thrombocytopenia, 9.3% had acute renal failure, 8.1% had fever, 8.1% had conjunctivitis, 7.6% had septicemia, and 5.2% had sepsis.

In multivariable survey logistic regression models including age, race/ethnicity, and sex as covariables, measles was associated with numerous organ-specific complications: gastrointestinal (aOR [95% CI] for dehydration: 3.96 [2.63–5.96]; diarrhea: 8.18 [5.12–13.05]; hepatitis: 4.37 [1.65–11.55]), hematologic (pancytopenia: 5.98 [2.64–13.57]; thrombocytopenia: 3.88 [2.32–6.51]), infectious (fever: 2.29 [1.13–4.65]; sepsis/SIRS: 2.73 [1.51–4.92]; septicemia 3.15 [1.88–5.28]), neurologic (encephalitis: 39.84 [16.51–96.12]; meningitis: 4.11 [1.32–12.80]),

**Association of Measles with Organ-Specific Complications**

| Complications | Measles | | | | | | |
|---|---|---|---|---|---|---|---|
| | **No** | **Yes** | | | | | |
| | Wtd Freq (%) | Wtd Freq (%) | Crude OR [95% CI] | P-value | Adjusted OR [95% CI] | P-value | |
| **Gastrointestinal** | | | | | | | |
| Dehydration | 27,650,224 (5.9%) | 161 (15.8%) | 2.99 [2.06-4.35] | <0.0001 | 3.96 [2.63-5.96] | <0.0001 | |
| Diarrhea | 7,937,366 (1.7%) | 97 (9.5%) | 6.12 [3.88-9.66] | <0.0001 | 8.18 [5.12-13.05] | <0.0001 | |
| Enterocolitis | 6,125,651 (1.3%) | 28 (2.7%) | 2.21 [1.02-4.82] | 0.045 | 2.15 [0.95-4.84] | 0.065 | |
| Hepatitis | 2,483,470 (0.5%) | 24 (2.4%) | 4.61 [1.95-10.87] | 0.001 | 4.37 [1.65-11.55] | 0.003 | |
| Nausea/Vomiting | 10,218,733 (2.2%) | 35 (3.5%) | 1.62 [0.76-3.43] | 0.210 | 1.75 [0.82-3.73] | 0.146 | |
| Pancreatitis | 7,647,125 (1.6%) | ≤10 (0.5%) | 0.33 [0.06-1.89] | 0.211 | 0.34 [0.06-2.03] | 0.234 | |
| **Hematologic** | | | | | | | |
| Pancytopenia | 2,698,812 (0.6%) | 35 (3.4%) | 6.20 [2.94-13.09] | <0.0001 | 5.98 [2.64-13.57] | <0.0001 | |
| Thrombocytopenia | 13,097,374 (2.8%) | 97 (9.5%) | 3.42 [2.11-5.54] | <0.0001 | 3.88 [2.32-6.51] | 0.438 | |
| **Infectious / Other** | | | | | | | |
| Cellulitis | 19,082,604 (4.1%) | 42 (4.2%) | 1.57 [0.76-3.25] | 0.222 | 1.05 [0.52-2.15] | 0.886 | |
| Fever | 4,802,035 (1.0%) | 73 (7.2%) | 7.68 [4.50-13.10] | <0.0001 | 2.29 [1.13-4.65] | 0.022 | |
| Sepsis/SIRS | 16,259,512 (3.5%) | 64 (6.3%) | 1.89 [1.09-3.31] | 0.025 | 2.73 [1.51-4.92] | 0.001 | |
| Septicemia | 20,204,516 (4.3%) | 84 (8.3%) | 2.04 [1.25-3.33] | 0.004 | 3.15 [1.88-5.28] | <0.0001 | |
| **Neurologic** | | | | | | | |
| Encephalitis | 250645 (0.1%) | 24 (2.4%) | 46.77 [19.63-111.44] | <0.0001 | 39.84 [16.51-96.12] | <0.0001 | |
| Meningitis | 976,510 (0.2%) | 14 (1.3%) | 6.72 [2.25-20.11] | 0.001 | 4.11 [1.32-12.80] | 0.015 | |
| **Ophthalmologic** | | | | | | | |
| Conjunctivitis | 1,064,164 (0.2%) | 87 (8.5%) | 41.48 [26.03-66.08] | <0.0001 | 27.20 [10.90-50.14] | <0.0001 | |
| Keratitis | 82,602 (0.02%) | ≤10 (0.5%) | 28.38 [4.107-196.10] | 0.0007 | 34.86 [5.19-234.24] | 0.0003 | |
| **Pulmonary** | | | | | | | |
| Bronchitis | 5,422,706 (1.2%) | 49 (4.8%) | 4.33 [2.30-8.14] | <0.0001 | 1.31 [0.49-3.55] | 0.594 | |
| Otitis Media | 1,752,201 (0.4%) | 37 (3.7%) | 10.14 [5.07-20.28] | <0.0001 | 3.17 [1.43-7.02] | 0.004 | |
| Pleurisy | 9,186,882 (2.0%) | 38 (3.8%) | 2.01 [1.01-4.02] | 0.048 | 3.01 [1.41-6.44] | 0.004 | |
| Pneumonia (all) | 37,095,861 (7.9%) | 127 (12.5%) | 1.66 [1.09-2.53] | 0.017 | 2.27 [1.43-3.58] | 0.001 | |
| Pneumonia (bacterial) | 19,062,938 (4.1%) | 79 (7.8%) | 2.04 [1.24-3.36] | 0.005 | 3.06 [1.73-5.42] | 0.0001 | |
| Pneumonia (viral) | 391,381 (0.1%) | ≤10 (0.9%) | 11.38 [2.90-44.72] | 0.001 | 3.83 [1.01-14.67] | 0.049 | |
| **Renal** | | | | | | | |
| Acute Renal Failure | 36,439,508 (7.8%) | 106 (10.4%) | 1.44 [0.95-2.20] | 0.089 | 2.65 [1.64-4.26] | <0.0001 | |
| Hypocalcemia | 2,930,458 (0.6%) | 24 (2.4%) | 4.15 [1.79-9.61] | 0.0009 | 5.17 [2.15-12.41] | 0.0002 | |
| Hyposmolality / Hyponatremia | 24,089,554 (5.2%) | 145 (14.3%) | 3.08 [2.09-4.54] | <0.0001 | 5.03 [3.34-7.59] | <0.0001 | |
| Urinary Tract Infection | 38,469,041 (8.2%) | 82 (8.1%) | 0.78 [0.38-1.58] | 0.490 | 2.27 [1.34-3.84] | 0.002 | |

Adjusted OR [95%CI]

**Fig 2. Association of measles with organ-specific complications.** Frequencies and multivariable logistic regression models showing association of measles with gastrointestinal, hematologic, infectious, neurologic, ophthalmologic, pulmonary, and renal complications.

ophthalmologic (conjunctivitis: 27.20 [10.90–50.14]; keratitis: 34.86 [5.19–234.24]), pulmonary (otitis media: 3.17 [1.43–7.02]; pleurisy: 3.01 [1.41–6.44]; pneumonia of any cause: 2.27 [1.43–3.58]; bacterial pneumonia: 3.06 [1.73–5.42]; viral pneumonia: 3.83 [1.01–14.67]), and renal (acute renal failure: 2.65 [1.64–4.26]; hypocalcemia: 5.17 [2.15–12.41]; hyponatremia: 4.75 [2.59–8.72]) (Fig 2). Enterocolitis (2.21 [1.02–4.82]) and bronchitis (4.33 [2.30–8.14]) showed higher odds only in bivariable models.

## Length of stay, admission, disposition and mortality

The mean [95% CI] LOS was significantly higher in patients with vs. without measles (4.8 [4.4–5.4] vs. 4.5 [4.4–4.5], P<0.01). Mean LOS increased slightly over time (Fig 3A) and was generally higher in older patients (Fig 3B). The strongest association with increased LOS was ≥2 chronic conditions (linear regression; adjusted beta [95% CI]: 0.76 [0.46–1.06], P<0.0001) (S2 Table).

Inpatients with measles most frequently were admitted from the emergency room (51.6% [39.7–63.6%]) or referral from a physician, outpatient center or clinic (41.4% [31.2–55.6%]). When stratified by race/ethnicity, a lower proportion of whites vs. non-whites were admitted from the emergency room (white: 50.2%; black: 100.0%; Hispanic: 66.7%; Asian/Pacific Islander: 61.7%), while a higher proportion of whites vs. non-whites were admitted from referral from a physician, outpatient center or clinic (white: 49.8%; black: 0.0%; Hispanic: 33.3%; Asian/Pacific Islander: 38.3%). When stratified by age, admission from the emergency room was generally higher with increasing age (0–9: 40.2%; 10–19: 50.0%; 20–39: 79.8%; ≥40: 51.7%) and admission from referral by a physician, outpatient center or clinic was generally lower with increasing age (0–9: 55.7%; 10–19: 0.0%; 20–39: 20.2%; ≥40: 44.0%).

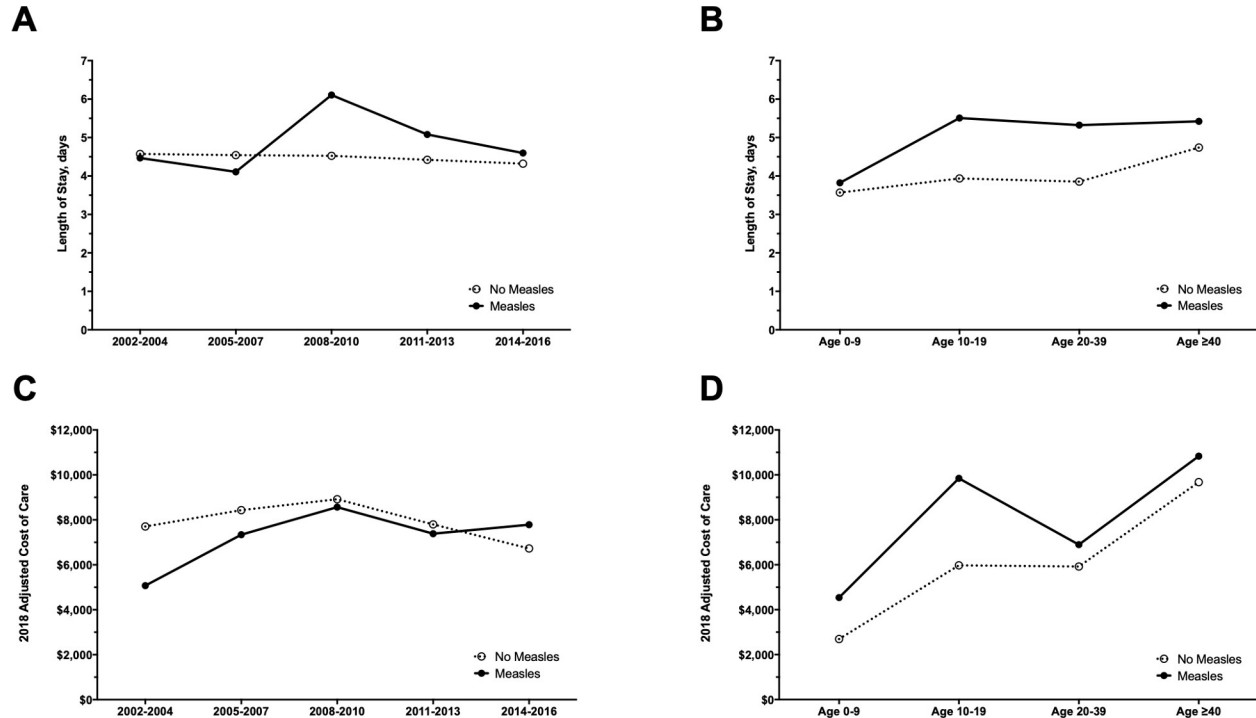

**Fig 3. Length of stay and cost of care.** Length of stay by year (A) and age (B) and cost of care by year (C) and age (D) for hospitalized patients with and without measles.

Inpatients with measles were most frequently routinely discharged to home or other self-care (84.3% [79.3–89.2%]), followed by transfer to other facilities (e.g. skilled nursing facilities, immediate care facilities) (8.5% [4.7–12.3%]). Inpatient mortality was not significantly higher in those with vs. without measles (proportion ± SEM: 3.3±1.2% vs. 2.3±0.01%, P = 0.333).

### Cost of inpatient care

The mean [95% CI] inflation-adjusted cost of inpatient care was not significantly different for those with vs. without a diagnosis of measles $7,438 [$6,446-$ 8,582] vs. $7,854 [$7,774-$7,935], P>0.05) and increased over time (Fig 3C). Mean costs generally increased with age in inpatients with and without measles (Fig 3D). The annual mean cost of measles hospitalization was $1,131,586 (range: $207,249 -$3,444,708) resulting in a total cost of $16,973,795 [$11,628,652-$22,318,934] from 2002–2016. Positive predictors of cost included ≥2 chronic conditions (linear regression; adjusted beta [95% CI], P-value; 0.73 [0.34, 1.12], P = 0.0003) and increasing length of stay (2 days: 0.58 [0.01–1.14], P = 0.046; ≥3 days: 1.23 [0.72–1.74], P<0.0001), while negative predictors of cost included decreasing urban population (not metropolitan or micropolitan: -1.11 [-2.07, -0.15], P = 0.021) (S3 Table).

## Discussion

This study found increasing hospitalizations for measles in the US between 2002–2016, with prolonged and costly hospitalizations. The estimated total number of measles hospitalizations in the US from 1977–1984 was 13,710 [26], which increased from 1985–1996 to 28,047, driven largely by the 1989–1991 epidemic [27]. From 1996 to 2002 measles hospitalizations were at an all-time low in US history with an estimated ≤23 cases annually [27]. While more recent data is lacking, an analysis of measles hospitalizations in a single children's hospital in Minnesota from 2011–2017 showed 33 total cases, driven largely by a 2017 outbreak in the Somali-American community [30]. Despite measles being officially declared eliminated from the US in 2000 [23,24], this study found 1,018 weighted hospitalizations for measles (mean of 68 cases per year) and rising number of measles hospitalizations from 2002–2016.

Increasing measles hospitalization mirrors a resurgence in measles cases nationwide. The frequency of documented measles cases from 2001–2015 was estimated to be 1,789, resulting in a low (<1 per million persons) but climbing incidence over time [31]. Based on official case reporting to the CDC and its National Notifiable Diseases Surveillance System (NNDSS), from 2001–2008 there were 557 cases of measles resulting in an estimated 126 hospitalizations, which increased in 2009–2014 with 1,264 cases and an estimated 211 hospitalizations [32]. From 2002–2016, publicly available data from NNDSS showed a total of 1,985 confirmed (i.e., laboratory-confirmed or epidemiologically linked cases) measles cases nationwide. Our analysis of the NIS during the same time period showed 1,014 weighted hospitalizations–a higher proportion of hospitalizations to cases than the CDC has estimated based on historical data (1 in 4 cases). The NNDSS is a passive reporting system that receives voluntary reports from state public health departments. Whereas, the NIS relies on discharge diagnoses coded as ICD-9 and ICD-10 data. Due to the passive nature of NNDSS reporting, previous studies suggested that reporting may be incomplete [32], particularly with hospitalized patients, as has been observed in past outbreaks [33]. Similar findings were seen internationally with other passive reporting systems [34]. Conversely, weighted discharge diagnoses using ICD-9 and ICD-10 codes may slightly overestimate measles cases in inpatients, as some cases with a measles diagnosis may have been rule-out diagnoses or misdiagnoses. Though, this is less likely since they are diagnoses provided at discharge that report the final diagnoses of those hospitalizations. Regardless, our results suggest there is increasing hospitalization for measles in the US over time.

A common finding in post-elimination years was that >80% of cases were in persons who were unvaccinated or had unknown vaccination status, and >60% of cases with philosophical or religious objection to vaccination [32], a theme echoed in other studies [35,36]. Decreasing incidence with age and decreasing proportion of imported and vaccinated cases suggests failure to vaccinate, rather than vaccine failure, was the driving force for the rising cases of measles in the US [31].

Measles can be perceived as a benign illness with no major consequences. However, as measles cases reemerge in the US, so do disease complications requiring hospitalization. We found gastrointestinal, hematologic, infectious, neurologic, ophthalmologic, pulmonary, and renal complications in inpatients with measles. From 1977–1984, the most common complications among hospitalized patients included pneumonia or other respiratory complications (34%), otitis media (8.5%), and encephalitis/convulsions/coma (3.4%); while pneumonia and otitis media frequency decreased with age, neurologic complications increased [26]. From 1987–2000, among all cases of measles, diarrhea (8.2%), otitis media (7.3%), pneumonia (5.9%), and encephalitis (0.1%) were the most prevalent [37]. From 2009–2014 complications were seen in 9% of cases, most commonly diarrhea (38%), pneumonia (33%), dehydration (25%), otitis media (16%), thrombocytopenia (10%), encephalitis (3%), pancytopenia (1%), and hepatitis (1%) [32]. We similarly found increased frequencies of these conditions, with dehydration, hyponatremia, pneumonia (primary or secondary bacterial/viral), acute renal failure, diarrhea, and thrombocytopenia showing highest prevalence among inpatients. Interestingly, keratitis was extremely rare. The rarity of keratoconjunctivitis may be explained by low rates of vitamin A deficiency in the US. Encephalitis showed the strongest positive association for measles diagnosis, with nearly 40-fold increased odds. Neurologic complications are associated with the most chronic morbidity. Post-measles encephalomyelitis can occur in 1 per 1,000 cases. Less commonly, measles inclusion body encephalitis or subacute sclerosing panencephalitis can present years after acute measles infection [22]. Previous studies showed that risk factors for severe illness include age <5 years and adults >20 years, immunodeficiency, pregnancy, malnutrition, crowding, and vitamin A deficiency [38].

Inpatient mortality was 3.3% (34 deaths) among measles inpatients, which was numerically higher than other inpatients (2.3%) and nearly 10-fold higher than overall mortality estimates of all cases of measles in the US (1 in 1,000). Hospitalized patients likely had more severe measles with complications and worse outcomes. The inpatient deaths observed in NIS are higher than reports of either no verified deaths from measles in NNDSS from 2009–2014 or one verified measles-related death in 1993–2002 from state death certificate data and the National Immunization Project that relies on direct reporting of states to the CDC [39]. Each of these reporting systems is limited by guidelines for measles case reporting (e.g. laboratory confirmation for CDC reporting), coding of death certificate diagnoses (limited to an acute and/or single direct cause of death), and state to federal communication. These limitations may result in underestimates of the true mortality of measles in the US [39].

Measles hospitalizations were more likely in non-white race/ethnicity, including Asians/Pacific Islanders, Hispanics, and Native American/others. Racial/ethnic disparities in measles vaccine coverage were thought to be reduced and/or potentially eliminated following targeted interventions by the CDC after the US measles epidemic of 1989–1991 [40]. However, there may be ongoing racial/ethnic differences in access to care, health literacy, and socioeconomic status driving the increased hospitalization observed in this study. This is supported by the finding that Asians/Pacific Islanders and Hispanics who were hospitalized with measles were more likely to present to the ED than whites. Hospitalization for measles was inversely associated with increasing age, which mirrored ours and others findings of decreased incidence with increased age [31]. While measles is thought to affect males and females similarly, our NIS

data and nationwide CDC measles prevalence data from 2001–2015 (52.9% male) suggest a slight male predilection in the US [31]. While some have observed higher complication and mortality rates in females vs. males [41], others have found no significant difference [42].

The average cost per measles-related hospitalization was $7,438 with total cost from 2002–2016 of $16,973,795. There was an estimated $10,000 cost per hospitalization during the 1989–1991 US measles outbreak [11], and while recent cost estimates are limited, the median cost per measles hospitalization in a single children's hospital from 2011–2017 was $5,291 [30]. Measles hospitalization costs include infection control required to stop nosocomial spread, such as isolation precautions, testing of exposed individuals, vaccine/immunoglobulin administration, tracing of cases, and the personnel hours required to carry out these tasks [43,44]. These represent the direct costs incurred in the inpatient setting, but do not capture indirect patient- and family-costs of measles hospitalizations. These costs also do not reflect the broader public health costs of measles, related to thorough case investigation, tracing of contacts to ascertain an index case, communication with the public, coordination between local, state, and federal health organizations, isolation and testing of exposed individuals, and post-exposure prophylaxis, which have been estimated to cost as high as $181,679 [45,46]. The public health cost of 16 measles outbreaks with 107 confirmed cases in the US in 2011 was estimated to be $2.7-$5.3 million [47]. Vaccination is both the most effective and cost-effective solution to prevent the spread of measles [48].

Study strengths include analysis of a nationally representative sample of all-payer data over a period of 15 years with over 96 million records. These results depict the serious and debilitating complications of measles infections resulting in hospitalization. However, milder measles cases are likely not captured in NIS. Limitations of this study include the lack of data on vaccination, serology or severity for measles, or inpatient treatment. Rare encephalitic complications can present years after measles and may be difficult to dissociate from more acute measles presentations. Identification of measles was performed using ICD-9-CM and ICD-10-CM codes and not verified by review of the health record or cross-referencing with measles registries. This could result in overestimation of the frequency and mortality rate of measles hospitalizations compared to expected estimates. However, recent studies suggest that ICD codes are effective in capturing several reportable diseases including measles and have improved over time [49].

The measles vaccine is inexpensive, extremely effective, and potentially lifesaving. The CDC estimates that among US children born from 1994–2013, the measles vaccine will prevent 322 million cases, 21 million hospitalizations, and 732,000 deaths over the course of their lifetimes, resulting in a savings of $295 billion in direct costs and $1.38 trillion in societal costs. More research and nationwide vaccination campaigns are needed to address measles resurgence in the US.

## Supporting information

**S1 Table. ICD-9-CM and ICD-10-CM codes used to identify measles and complications.**
(DOCX)

**S2 Table. Predictors of length of stay in patients with measles.**
(DOCX)

**S3 Table. Predictors of cost of care in patients with measles.**
(DOCX)

## Author Contributions

**Conceptualization:** Jonathan I. Silverberg.

**Data curation:** Raj Chovatiya, Jonathan I. Silverberg.

**Formal analysis:** Raj Chovatiya, Jonathan I. Silverberg.

**Funding acquisition:** Raj Chovatiya, Jonathan I. Silverberg.

**Investigation:** Raj Chovatiya, Jonathan I. Silverberg.

**Methodology:** Raj Chovatiya, Jonathan I. Silverberg.

**Project administration:** Raj Chovatiya, Jonathan I. Silverberg.

**Resources:** Raj Chovatiya, Jonathan I. Silverberg.

**Software:** Raj Chovatiya, Jonathan I. Silverberg.

**Supervision:** Raj Chovatiya, Jonathan I. Silverberg.

**Validation:** Raj Chovatiya, Jonathan I. Silverberg.

**Visualization:** Raj Chovatiya, Jonathan I. Silverberg.

**Writing – original draft:** Raj Chovatiya, Jonathan I. Silverberg.

**Writing – review & editing:** Raj Chovatiya, Jonathan I. Silverberg.

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
