## [Decision Letter · Decision Letter 0]

22 Jan 2020

PONE-D-19-30745

Inpatient morbidity and mortality of measles in the United States

PLOS ONE

Dear Dr Silverberg,

Thank you for submitting your manuscript to PLOS ONE. After careful consideration, we feel that it has merit but does not fully meet PLOS ONE’s publication criteria as it currently stands. Therefore, we invite you to submit a revised version of the manuscript that addresses the points raised during the review process.

We would appreciate receiving your revised manuscript by Mar 07 2020 11:59PM. To enhance the reproducibility of your results, we recommend that if applicable you deposit your laboratory protocols in protocols.io, where a protocol can be assigned its own identifier (DOI) such that it can be cited independently in the future. For instructions see: http://journals.plos.org/plosone/s/submission-guidelines#loc-laboratory-protocols

We look forward to receiving your revised manuscript.

Kind regards,

Ka Chun Chong

Academic Editor

PLOS ONE

Reviewers' comments:

Reviewer's Responses to Questions

**Comments to the Author**

1. Is the manuscript technically sound, and do the data support the conclusions?

Reviewer #1: Yes

Reviewer #2: Yes

2. Has the statistical analysis been performed appropriately and rigorously? 

Reviewer #1: I Don't Know

Reviewer #2: Yes

3. Have the authors made all data underlying the findings in their manuscript fully available?

Reviewer #1: No

Reviewer #2: Yes

4. Is the manuscript presented in an intelligible fashion and written in standard English?

Reviewer #1: Yes

Reviewer #2: Yes

5. Review Comments to the Author

Reviewer #1: REVIEWERS COMMENTS PLOSONE JANUARY 2020

(PONE-D-19-307-45)

Reviewer’s Report

Title: Inpatient Morbidity and Mortality of Measles in the United States

Date: 11.01.2020

Version 1

General comments: Topic is of relevance and global interest. Reads well but introduction and methodology do not contain sufficient detail to appreciate study and anticipate results.

Title: Title seems appropriate. Suggest adding source of data (discretionary)

Abstract:

Page 3 line 48-49, 52-53: A concise summary but more information on background and analysis would be useful.

Page 3: Line 62; suggest writing defining vs by also writing it in full and defining other abbreviations when first mentioned.

Introduction:

Page 4 line 84: suggest removing “that”

Introduction

Suggest adding additional information such as - Information on the epidemiology of the most recent outbreaks in the US, any changes in predictors of hospitalisation in the US over the years or experience from other developed countries - useful for understanding of the context surrounding this study and changes to expect.

Methods:

Study area: No detail. It would be useful to know the geographic distribution of hospitals included in the sample, rural -urban differences etc whether it cover high risk locations e.g. migrant populations?

Page 6 line 97- Sampling: Suggest more detail is provided on the sampling procedure – what criteria informs the weighting? How many hospitals were represented in this sample? What types of hospitals were included, how many were children’s hospital’s?

Did the sample include hospitals from communities that had an outbreak during the period and were these populations targeted?

Study Population: What were the inclusion criteria- ICD 9/10 criteria alone? What proportion had laboratory confirmation? The age of patients was not specified? Were there any other exclusion criteria?

Data collection

Page 6: line 103- 10: What information was collected from the database? Though some is reported with the statistical analysis, but this aspect is not clear. It will be helpful to know the information collected on patient level characteristics, hospital level characteristics and outcome measures? Was there collection of data on immunisation status?

Page 6 line 106-7: Provide more detail on the co-morbidities studied especially those known to be associated with hospitalisations from measles what proportion had chronic respiratory conditions? Was it a risk factor for hospitalisations and longer duration of stay? Was this analysis done?

Results

Page 9 line Table 1 what chronic conditions?

Page 10 line 153 – suggest “Factors associated with hospitalization for measles”

Page 10: line 164 – “complications” kindly rephrase and make it clearer .

Page 10 line 169 - any sex differences?

Page 13 line 192 and 217 also need to be rephrased.

Page13 line 208; Define abbreviation ED

Why were children <5 years not considered as a separate group in the analysis?

Discussion

Extensive and covers findings

Page 17 line 287: contains mortality data not seen in results

Limitation Page 19 lines 331 and 332 should have also been in the methods section.

Conclusion

Section for conclusion not highlighted and should relate to objectives and findings of study as done in the abstract.

Graphs: some graphs do not have titles

References:

Page 29 line 366-369 in capitals – any reason?

Suggest including more recent articles.

Reviewer #2: This MS captures topical and relevant data from a nationwide sample in US. Although it does not cover the most recent period, it should be of use to policy makers and program managers to make informed decisions about impact of renewed measles transmission in US.

The analysis is straightforward and logical. And the language easy to follow. I have made a coupe of minor comments to sharpen some points.

6. PLOS authors have the option to publish the peer review history of their article (what does this mean?). If published, this will include your full peer review and any attached files.

Reviewer #1: No

Reviewer #2: Yes: Anindya Sekhar Bose

---

## [Author Response · Author response to Decision Letter 0]

31 Jan 2020

To Whom It May Concern:

Thank you very much for the helpful reviewers’ comments for our manuscript. We have addressed these comments as follows.

Reviewer #1: 

• Page 3 line 48-49, 52-53: A concise summary but more information on background and analysis would be useful.

o We have updated the background and methods section of our abstract to provide further detail for the reader.

• Page 3: Line 62; suggest writing defining vs by also writing it in full and defining other abbreviations when first mentioned.

o We have updated the abstract to define any abbreviations

• Page 4 line 84: suggest removing “that”

o We have incorporated this suggestion into the manuscript

• Introduction: Suggest adding additional information such as - Information on the epidemiology of the most recent outbreaks in the US, any changes in predictors of hospitalisation in the US over the years or experience from other developed countries - useful for understanding of the context surrounding this study and changes to expect.

o We appreciate the suggestion to further discuss the epidemiology of outbreaks in the US, as well as predictors of hospitalizations for further context. For the sake of brevity and clarity, this information is already discussed in detail in the Discussion section and compared with the data gathered in this study. In particular, we discuss the estimated numbers of measles hospitalizations in the preceding decades, estimated numbers of measles cases and hospitalizations since US elimination, complications associated with measles hospitalizations in the preceding decades, and previously characterized risk factors for hospitalization. There is limited demographic data available in previous studies to compare with some of our findings as shown in Table 1. We have limited our main focus to inpatient measles hospitalizations as opposed to measles outbreaks in general, as this is an equally important but distinct topic from our focus. Nevertheless, since measles elimination in the US, as mentioned in the Introduction, US outbreaks have resulted from travel-related exposures and communities with low rates of vaccination, and the number of hospitalized patients have correlated with the total number of cases in any given year.

• Methods: 

Study area: No detail. It would be useful to know the geographic distribution of hospitals included in the sample, rural -urban differences etc whether it cover high risk locations e.g. migrant populations? 

Page 6 line 97- Sampling: Suggest more detail is provided on the sampling procedure – what criteria informs the weighting? How many hospitals were represented in this sample? What types of hospitals were included, how many were children’s hospital’s?

 Did the sample include hospitals from communities that had an outbreak during the period and were these populations targeted?

o We have added additional information about the NIS dataset and sampling procedure to the Methods section. The NIS provides a 20% cross-sectional sample of all hospitals in the US, regardless of rural/urban, childrens/general, etc.

• Study Population: What were the inclusion criteria- ICD 9/10 criteria alone? What proportion had laboratory confirmation? The age of patients was not specified? Were there any other exclusion criteria?

o As discussed in the Methods section, measles and various comorbidities were specified with ICD9 and ICD10 criteria. As the NIS is a nationwide cross-sectional database, it does not include information regarding treatment, lab testing, vaccination, and other specific data found in the patient medical record. This is listed as a limitation of the study in the discussion section. The NIS includes patients of all ages without any exclusions criteria, as previously explained, and age breakdowns for measles patients are further discussed in the Results section.

• Page 6: line 103- 10: What information was collected from the database? Though some is reported with the statistical analysis, but this aspect is not clear. It will be helpful to know the information collected on patient level characteristics, hospital level characteristics and outcome measures? Was there collection of data on immunisation status?

o Baseline characteristics about patient demographics and hospital information obtained from the NIS are summarized in Table 1. Further details about outcomes including death and discharge are discussed in the Length of Stay, Admission, Course, Disposition and Mortality subsection in the Results. As previously mentioned, the NIS is a weighted cross-sectional database that does not include laboratory, treatment, or immunization data. 

• Page 6 line 106-7: Provide more detail on the co-morbidities studied especially those known to be associated with hospitalizations from measles what proportion had chronic respiratory conditions? Was it a risk factor for hospitalisations and longer duration of stay? Was this analysis done?

o The wording of the Methods section was updated to reflect our analysis of complications rather than comorbidities. An important aim of the study was to better understand measles-related complications in inpatients. Table 2, Supplementary Table 1, and the Complications subsection of Results summarize the complications associated with measles hospitalizations. Specific long-standing comorbidities can be more difficult to accurately assess in the NIS if they are not directly relevant to the specific encounter. However, a variable that measures the overall number of chronic comorbid conditions is included in the NIS as mentioned in the Methods section. We discuss in the Results section that increasing number of chronic conditions was inversely associated with measles hospitalization but positively associated with length of stay.

• Page 9 line Table 1 what chronic conditions?

o The NIS includes a data entry that contains the number of chronic diagnoses (condition ≥ 12 months) that meets specific criteria regarding limitations on independence and need for ongoing care. Additional clarification has been provided in the Methods section.

• Page 10 line 153 – suggest “Factors associated with hospitalization for measles”

o We appreciate the suggestion and have updated the wording.

• Page 10: line 164 – “complications” kindly rephrase and make it clearer .

o We appreciate the suggestion and have updated the wording.

• Page 10 line 169 - any sex differences?

o No sex differences were seen. This was not a focus of this particular analysis for the manuscript.

• Page 13 line 192 and 217 also need to be rephrased.

o We appreciate the suggestion and have updated the wording

• Page13 line 208; Define abbreviation ED

o We appreciate the suggestion and have updated the wording

• Why were children <5 years not considered as a separate group in the analysis?

o Our purpose in stratifying admission source (emergency room, physician’s office) by age was to show broad trends among children, adolescents, young adults, and older adults. The cutoffs of age selected for the younger age groups corresponded to WHO definitions of childhood (under 10 years) and adolescence (over 10 years)

• Page 17 line 287: contains mortality data not seen in results

o We kindly refer to you Table 1 and the last section of the Results which discuss both the number of deaths among measles inpatients as well as the mortality rate.

• Limitation Page 19 lines 331 and 332 should have also been in the methods section.

o We have updated the methods section with this information

• Graphs: some graphs do not have titles

o We appreciate the suggestion and have updated the titles.

• Page 29 line 366-369 in capitals – any reason?

o Capitols have been removed as this was a formatting error from the citation software.

Reviewer #2

• Overall comments: This MS captures topical and relevant data from a nationwide sample in US. Although it does not cover the most recent period, it should be of use to policy makers and program managers to make informed decisions about impact of renewed measles transmission in US. The analysis is straightforward and logical. And the language easy to follow. I have made a couple of minor comments to sharpen some points.

o We appreciate the reviewer’s comment.

• Pg 8 line 148 - Implications of different forms of medical/health insurance in the US should be added for an international audience

o We have clarified the definitions of Medicare and Medicaid in the methods section in order to better explain US insurance options for a foreign population.

• Pg 17 line 289 - This is not a valid comparison as hospitalized cases of measles are usually more severe and tend to have higher mortality than non-hospitalized cases.

o We understand that the inpatient mortality rate is not the same as overall mortality rate and will likely be higher based more severe cases. This is why the next sentence in this section explains that hospitalized patients likely had more severe cases of measles with worse outcomes, thus suggesting that you indeed cannot exactly equate inpatient mortality rate to overall mortality rate. However, this estimate of overall mortality (1 in 1,000) is one of the only estimates of mortality related to measles that has been published by the CDC, and this figure is routinely used by public health officials and the scientific community in the US. Our comparison here was simply to show that (1) our inpatient mortality rate appears higher than the overall measles mortality rate, (2) this is likely related to more severe inpatient cases with worse outcomes, and (3) there is a lack mortality data (both inpatient and in general) for measles in the US, particularly after elimination.

Thank you again for considering our manuscript for publication in PLOS One. All authors have read and approved the manuscript, its contents, and its submission to PLOS One.

Thank you very much.

---

## [Decision Letter · Decision Letter 1]

13 Mar 2020

PONE-D-19-30745R1

Inpatient morbidity and mortality of measles in the United States

PLOS ONE

Dear Dr Silverberg,

Thank you for submitting your manuscript to PLOS ONE. After careful consideration, we feel that it has merit but does not fully meet PLOS ONE’s publication criteria as it currently stands. Therefore, we invite you to submit a revised version of the manuscript that addresses the points raised during the review process.

We would appreciate receiving your revised manuscript by Apr 27 2020 11:59PM. To enhance the reproducibility of your results, we recommend that if applicable you deposit your laboratory protocols in protocols.io, where a protocol can be assigned its own identifier (DOI) such that it can be cited independently in the future. For instructions see: http://journals.plos.org/plosone/s/submission-guidelines#loc-laboratory-protocols

We look forward to receiving your revised manuscript.

Kind regards,

Ka Chun Chong

Academic Editor

PLOS ONE

Additional Editor Comments (if provided):

This revised manuscript have addressed most of the reviewers' comments. Please address the last comments from the reviewers before a formal acceptance.

Reviewers' comments:

Reviewer's Responses to Questions

**Comments to the Author**

1. If the authors have adequately addressed your comments raised in a previous round of review and you feel that this manuscript is now acceptable for publication, you may indicate that here to bypass the “Comments to the Author” section, enter your conflict of interest statement in the “Confidential to Editor” section, and submit your "Accept" recommendation.

Reviewer #1: (No Response)

Reviewer #2: (No Response)

2. Is the manuscript technically sound, and do the data support the conclusions?

Reviewer #1: Yes

Reviewer #2: Yes

3. Has the statistical analysis been performed appropriately and rigorously? 

Reviewer #1: I Don't Know

Reviewer #2: Yes

4. Have the authors made all data underlying the findings in their manuscript fully available?

Reviewer #1: No

Reviewer #2: Yes

5. Is the manuscript presented in an intelligible fashion and written in standard English?

Reviewer #1: Yes

Reviewer #2: Yes

6. Review Comments to the Author

Reviewer #1: Reviewer’s Report version 2

Title: Inpatient Morbidity and Mortality of Measles in the United States

Date: 06.03.2020

Comments

The responses provided by the authors have largely addressed my comments. I have noted that the graphs are still not labelled, I mean, they do not have titles. I wonder if this can be addressed if the paper is to be published?

Reviewer #2: The authors have addressed most of the comments in the revised version. However, my earlier comments about inappropriate comparison between measles CFR has not been addressed.

The way the authors state this is as, "Inpatient mortality was 3.3% (34 deaths) among measles inpatients, which was numerically higher than other inpatients (2.3%) and nearly 10-fold higher than previous mortality estimates of all cases of measles in the US (1 in 1,000)." Stating that this estimate of measles related mortality is ten fold higher than "earlier estimates" seems to indicate that the authors have identified a new estimate measles CFR and is misleading as the two rates apply to different at risk populations.

Replacing 'previous' with 'overall' in quoted sentence should allow the reader to draw the correct conclusions about measles mortality. Barring this issue, I recommend that the MS should be accepted.

7. PLOS authors have the option to publish the peer review history of their article (what does this mean?). If published, this will include your full peer review and any attached files.

Reviewer #1: No

Reviewer #2: Yes: Anindya Sekhar Bose

---

## [Author Response · Author response to Decision Letter 1]

15 Mar 2020

To Whom It May Concern:

Thank you very much for the helpful reviewers’ comments for our manuscript. We have addressed these comments as follows.

Reviewer #1: 

• The responses provided by the authors have largely addressed my comments. I have noted that the graphs are still not labelled, I mean, they do not have titles. I wonder if this can be addressed if the paper is to be published?

o Each figure panel now has a label in the image, in addition to a figure title for each caption in the text

Reviewer #2

• The authors have addressed most of the comments in the revised version. However, my earlier comments about inappropriate comparison between measles CFR has not been addressed. The way the authors state this is as, "Inpatient mortality was 3.3% (34 deaths) among measles inpatients, which was numerically higher than other inpatients (2.3%) and nearly 10-fold higher than previous mortality estimates of all cases of measles in the US (1 in 1,000)." Stating that this estimate of measles related mortality is ten fold higher than "earlier estimates" seems to indicate that the authors have identified a new estimate measles CFR and is misleading as the two rates apply to different at risk populations. Replacing 'previous' with 'overall' in quoted sentence should allow the reader to draw the correct conclusions about measles mortality. Barring this issue, I recommend that the MS should be accepted.

o We appreciate the reviewer’s comment. We have updated the wording to reflect the more accurate comparison 

o We have added a data availability statement

---

## [Editor Report · Decision Letter 2]

23 Mar 2020

Inpatient morbidity and mortality of measles in the United States

PONE-D-19-30745R2

Dear Dr. Silverberg,

We are pleased to inform you that your manuscript has been judged scientifically suitable for publication and will be formally accepted for publication once it complies with all outstanding technical requirements.

With kind regards,

Ka Chun Chong

Academic Editor

PLOS ONE
---

## [Editor Report · Acceptance letter]

9 Apr 2020

PONE-D-19-30745R2 

Inpatient morbidity and mortality of measles in the United States 

Dear Dr. Silverberg:

I am pleased to inform you that your manuscript has been deemed suitable for publication in PLOS ONE. Congratulations! Your manuscript is now with our production department. 

With kind regards,

on behalf of

Dr. Ka Chun Chong 

Academic Editor

PLOS ONE